# The Politics and Aesthetic Choices of Feminist Art Criticism

**Katy Deepwell** 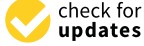

Department of Visual Arts, Middlesex University, London NW4 4BT, UK; k.deepwell@mdx.ac.uk

**Abstract:** This article explores feminist art criticism from the point of view of aesthetics/politics in global contemporary art. It is based on the author's experience as an art critic and founding editor of *n.paradoxa: international feminist art journal* (1998–2017). Reading articles published in the previous two decades both for the journal and outside it, it became possible to identify how subjects produce specific objects in art criticism that demonstrate different locations and standpoints in thought and how these align with criticism from broader feminist political theories. This is an exploration of the aesthetics/politics both in, about and beyond feminist art criticism. The methodology presented analyses feminist art criticism using a model of clusters of concepts that draws on Anne Ring Petersen's examination of identity politics, race and multiculturalism from 2012. Feminist analyses in which this approach has been attempted are discussed: Sue Rosser's 2005 analysis of cyberfeminism and Tuzyline Jita Allan's 1995 discussion of black/womanist/African feminisms. The article identifies four types of feminist art criticism: liberal feminism, materialist feminism, feminist cosmopolitan multi-culturalism, and queer post-colonial feminism. The aims, methods and approaches of these tendencies are outlined to demonstrate the differences between them. The article concludes with a discussion about the futures of feminist art criticism.

**Keywords:** feminist aesthetics; feminist politics; feminist art criticism; feminist art

This article explores feminist art criticism from the point of view of aesthetics/politics. Art criticism appears divided into "art writing" or "art theory". The first is seen as reactive to current exhibitions, artworks or artists and closer to journalism; the second is seen in largely pedagogical terms or as an academic exploration of theoretical concerns in contemporary art. In neither is it the case that discussing artworks produced by women artists (which manifest their own aesthetic qualities or might be identified as "feminist art") determines the type of art criticism written (as a political gesture, intervention or in the name of "feminism"). There exists much writing about women artists or feminist subjects in art, which is decidedly not sympathetic to feminisms, but is still about the art that women produced and some of this writing, especially when it is publicity, simply uses feminism as a label to promote work without serious attention to the content/context of feminist politics. The latter is known as "artworld feminism" and for many, it is "not-feminism".

Art criticism of contemporary art is always "after" the event/the exhibition/the work; its claims for "now" time, for a zeitgeist, for being in the moment or the latest phenomena, are always a reflection on what already exists–even when in journalism/art magazines, this time is shortened to a few hours, rather than the months of journal or book production. This trait it shares with philosophy in so far as it is a commentary on what already exists, especially previous texts. The reading may be productive in the future because it contains a possibility beyond the timeframe in which it was written. The negotiation between reading the artwork and relevant theories for a critic is the result of both political decision-making and aesthetic choices, even if this is understood as cultural politics with a small "p". How the subject/object of feminism appears in the text, therefore, represents a political/cultural choice by both critic in their relation to artwork and artist, but also one wherein the engagement/encounter with the artwork or artist mediates and challenges this relationship

as transparent or obvious, even where its time or its timeliness is not. How to trace and understand these choices as political/aesthetic ones is the subject of the article.

This negotiation is more than the decision about what type of art criticism to produce: in an autobiographical style or following the biography of an artist; naming an emerging type of artwork or written in a reflective/responsive, emotional or even poetic fashion–as indicated by textbooks about how to write art criticism today. The question of time in art criticism is more complex (because it follows art's production) than predicting or calling for the future to be different, characterising the past, or mediating the relationship of what is current to understanding the present. Feminism is a politics and it is not singular in its politics; as such, in its relation to art criticism, feminist art criticism also has to be understood as an umbrella term for diverse political/aesthetic approaches (feminisms) about how to think about art, artist and artworks. Given this potential for a range of feminist art criticisms, to speak of feminist art criticism in relation to questions of both the aesthetic and the political must involve more than simply advocacy for one position as the "true" or "representative" version of feminism. The tendency to advocate for a preferred feminism in art criticism is typically conducted as a claim for a "new" method for feminism in our time, while discounting or disregarding others in relation to the future as too essentialist, too limited, too centred on the wrong questions or issues, or just outdated. After 50 years of feminist art criticism and many articles, books, special issues and whole journals devoted to the subject, many ideological positions, tendencies and approaches have emerged. In what follows, I am deliberately not linking these approaches to individual authors as "their work", under a proper name, because I want to draw a picture of generic types, characterisations about the locations of arguments, positioning these in terms of standpoints that are ultimately ideological perspectives. Although this model was written with authors in mind–an "I" who writes–the instances described are those in which the text as a form/format writes itself, with the same repeated tropes present in many different authors.

The current context makes it hard to re-establish the relative importance of feminist readings in art criticism. The rhetoric of diversity and inclusion in contemporary universities, especially in the UK, has been openly hostile to feminisms, arguing it is superseded and no longer necessary (because women have equality in the West!), or that racial/ethnic diversities must take preference over all other differences, including black (as a political term) and/or lesbian feminist versions of feminisms. This article is not an argument for a "neutral" position nor is it a bird's eye overview that cancels out all other initiatives or possibilities, and it is written from my own perspective as a UK-based academic, i.e., what I read, see and find in the English language with which I work. I can, therefore, only identify how some forms of feminism operate. Although I am attempting here to name different types of feminisms in art criticism, identifying these approaches is not done as negative criticism, in the sense of negation to introduce another construction to the world, nor am I advocating a "new" position, by grounding my view in only a critique of "others". However, like all knowledge in the humanities, to write is always to take a partial and preferential view, to follow specific agendas in determining objects or subjects of study, and this is what I understand is meant about the epistemic violence of a text (to follow Gayatri Spivak) (Spivak 1985) because it has to be based on inclusions/exclusions, as it constructs its position.

Feminism in academia has been made to seem strangely anomalous, out of sync, too white, too liberal (i.e., too vague, open and tolerant, as opposed to neo-liberal), out-dated as "the past" not concerned with the present, peripheral in too many political agendas, and squeezed out of discourses as too difficult or too messy to deal with or impossible to grasp or understand as a position, except in a campaigning "me too" kind of way. It is tolerated as about advocacy for women's rights in non-Western parts of the world, but only as a developmental stage towards racial/gender equality. Identity politics–prioritising women's voices or experiences–has a large part to play as alibi to these false representations, as the default position to which feminisms are made to appeal or answer. These problems

in understanding or even tolerating feminisms are long-standing but neoliberalism has exacerbated them in its claim for supporting "diversity", but not feminist discourse, while ironically claiming to promote women's opportunities as leaders and/or middle managers, which does not change the position for the majority of working women (across races, classes, castes or religions). The censoring impact of social media also contributes in loud and noisy exchanges, as if feminism were only caught up in a competition between trans-people and biological women for limited space or an unrecognised right to exist, and neither were arguing for changes to improve people's lives. The nationalist right (most notably in Hungary and Russia) has done its best to shut down feminist discourse in women's studies in university. Art criticism is not a bell-weather for all these ills but set against these; the increasing volume of publishing by and about feminist art practices needs to be reconsidered as definitively against these kinds of prejudices and positions. Over 50 years, numerous publications that set out to correct, alter the account or produce new kinds of reading of art, and particularly that produced by women artists, in the name of feminism or as contributions to feminist debates, cannot so easily be dismissed. Analysing these approaches as a politics in reading artworks is worth the effort, if we are to uncover a different place for feminisms in art criticism of the global contemporary today.

In 2012, Anne Ring Petersen published a diagram of a "cluster of concepts" that showed different trajectories on 'cultural identity' in discourses about identity politics and institutional multi-culturalism (Petersen 2012). This model demonstrated how different interlinked concepts were reproduced in five lines of enquiry and "ways of doing" that predominated in art institutions, art history and art criticism. There are striking parallels here with how "feminisms" are handled as a rhetoric in the curating and criticism of contemporary art, as I will go on to suggest. In the article, Petersen was concerned with examining how inclusion/exclusion operated in the global art world, especially for non-Western artists, even when the rhetoric of a plural, inclusive and integrationist multi-culturalism was advanced and as identification of "cultural identity" repeatedly pointed to the influence of an individual's background on the production of their artworks. Her critique (and others) of the "integrationist" struggle experienced by Western institutions in a more internationalist context today was that many had resorted to recognising artists with migrant backgrounds living and working in the West, as well as non-Western artists as part of a global contemporary art world, only by re-centering Western concerns and privileging a version of multi-culturalism in which ethnic art, models of hybridisation and cultural difference rested on absolute identity markers for both artists and artworks. This re-centering was woven together by claims for their "authenticity", "identity" and "Otherness". Non-Western women artists have experienced this in their integration within mainstream institutions, but so have white Western women, where cases for their exceptionalism to the mainstream business of art have been made in the name of both "identity" and "Otherness". In the article, Petersen also offered a critique of the strategies in New Internationalism of the early 1990s, arguing that three conflicting aims (to this pattern of integration) were present: (1) to deconstruct the approaches of Western art institutions and art history for their ethnocentric and racist structures and practices while arguing for inclusion, (2) to pay greater attention to art's own significance, beyond a vehicle for identity politics or anthropological concerns and (3) at the same time, to emphasise complexity by maintaining a necessary focus on cultural differences and their relative importance. If one adds combating sexism, or gender/sexuality into Petersen's analysis, many of the same multi-faceted ambitions mark feminist concerns about women artists in the global art world.

The five lines of enquiry in Petersen's chart each begin with different "nodal points": on 'culture', 'ethnicity', 'migration', 'globalisation' or 'multiculturalism'–from which other sub-clusters point towards different priorities or areas of work (for example, in order) for 'the artworld', 'authenticity', 'transnational connections', 'cosmopolitanism' or 'the new internationalism/global art'. Through this chart, she was able to indicate how 'entangled, intersecting and antagonistic concepts' (Ibid., p. 200) reflect different agendas advanced

under the singular concept of cultural identity. She was able to show how even though there were many self-evident and recurrent binaries used: centre/periphery; national/foreign; Western/non-Western, more complex formulations than this are in operation and were articulated to include/exclude art practices or artists in terms of local vs. global; international vs global; transnational vs national; authentic traditional arts vs inauthentic/mimicry of Western art tropes; cosmopolitan vs regional modernisms. Her route out of these dilemmas, was first that we need to deconstruct the term "cultural identity" from its attachment to a single national/ethnic/racial identity; question the misconception that the "roots/routes" resulting from an artist's biography determine the work; and focus more on how the work's meaning is constructed by context of production/exhibition. As she puts it, we don't need "*more* identity politics, but rather a reconsideration of the works themselves from an aesthetic and epistemological point of view that does not lapse into old-fashioned modernist universalism and aestheticism" (Ibid., p. 204).

One strategy in response to Petersen's work might be to add "gender" to the cluster of concepts identified as it would reveal the differential position of non-Western women artists or women artists in the West with migrant backgrounds in her proposed model. A focus on gender and nationalisms might lead us to quite different constructions of knowledge in these clusters, dependent on the mediums used by the artists; the over-emphasis on "authentic/traditional/culturally-specific references" in non-Western women's work or a long list of other factors that combine to segregate women's perspectives on global issues from climate change to violence against women that perpetually disadvantage women on the wrong side of binary oppositions. We could also apply Chandra Mohanty's critique of Western research on women in the global South, in terms of who and how women are constructed as objects of other people's knowledge (Mohanty 1988). In reading women artists' works, it is always the case that gender tends to divide and reconstruct this kind of thinking on cultural identities differently–this is not different criteria, but shows their attention to difference within dominant value systems. Men's works also require feminist analysis according to their gendered perspectives and on more than grounds of masculinity–vulnerability, as well as male egotism–because gender is a binary term, not a signifier for only-women. Another strategy might be to ignore gender as a factor in race/class/nation/ethnicity analysis altogether as an additional complication too difficult to deal with and of secondary importance. There are plenty of examples of this!

Nevertheless, Petersen's method (drawn from sociology) prompted me to start to consider how clusters of concern and approaches to feminism (as if it were a singular concept) could be differentiated according to how women artists and their works were framed and objects for analysis set up. I looked across articles I had read as contributions to debates in feminist art in recent years or had chosen to publish in *n.paradoxa: international feminist art journal*, 1998–2017. In the following chart, I identified four positions as recognisable responses to the situation of contemporary art after 50 years of feminist interventions amongst artists, critics and curators in an ever-changing object called feminism.[1] While I did reflect on two decades running *n.paradoxa*, I also commissioned many other kinds of art writing that did not conform to the proposed model of articles: artists' interviews, analyses of an art scene in a country or region, extended exhibition reviews focused on a curator's approach, manifestos and artists' pages, as well as theoretical essays that aimed to set up new kinds of thinking about a theme.

In the chart I devised, I chose to name these four approaches as their advocates have named themselves, but they follow paths commonly identified by many feminists as different approaches to feminisms. The same distinction between liberal, socialist, radical, postmodern and post-colonial feminists has been deployed to examine culture as well as feminist politics and human nature (Jaggar 1983; Tong 1988). Debates about the political differences between liberal, socialist, radical/anarchist feminist, black feminist, indigenous feminist or lesbian feminist have been highlighted in many feminist studies, precisely because they lead to different political programs. The variety of approaches to this consider generational patterns, political visions for change and the limitations that approaches

demonstrate in their priorities and through different categories for thought. This is where politics/aesthetics enters the frame as pre-determining a set of choices about what "ought to be done" in feminist art criticism, an ideological effect within the construction of knowledge, effectively determining/over-determining standpoints and paradigms.

Sue Rosser's analysis of women's work in/against/for information technology (Rosser 2005), for example, successfully argues how the standpoints of liberal, socialist, radical, postmodern and post-colonial political approaches within feminism, each offer different analyses in how to think about information technology workforces, designs of systems and users/uses. She concludes and extends her argument by a discussion of emergence of cyberfeminism and tries to identify how these earlier approaches are also present in and through cyberfeminism's diverse existentialist, psychoanalytic, queer, transgender and postcolonial approaches to feminisms. While her approach to map these debates regarding computing via larger feminist politics can classify or identify differences in perspectives and the ideologies underpinning them, it is not the case that each standpoint necessarily builds on the critiques of the others in an evolutionary chain, superseding the other through rational argument. These approaches co-exist in the present amongst different women, even when the theories with which they engage may have developed at different times or have different routes or affiliations to other bodies of knowledge. However, this approach does bring clarity to political differences in how feminisms operate, but does it go beyond naming these tendencies or classifying people within these categories? Adorno's unfinished introduction to *Aesthetic Theory* offers a salutary warning to the task of trying to fashion any aesthetic theory–even a feminist aesthetics–as a complete system, and at the same time highlights why all value judgements should continue to name their presuppositions and approaches to knowledge production or systems of knowing. He identified the many incomplete and unfinished theories in aesthetics, highlighting their lack of coherence or system-building capability because, on the one hand, there existed the "fundamental difficulty, indeed impossibility, of gaining access to art by means of philosophical categories, and on the other, the fact that aesthetic statements have traditionally presupposed theories of knowledge" (Adorno 1997). In spite of the noted philosophical turn towards understanding contemporary art in the last decade, these reservations remain in place. My aim in proceeding with this rather abstract and formalist method was to think about the efficacy or even the effectiveness of certain feminist ways of framing their subject (artworks by women) and becoming more aware of their pre-suppositions and limits, in relation to theories of knowledge production. While new works of art will always confound and produce the necessity for revision of aesthetic categories, the expectations, claims and models erected to understand aesthetic/political issues cannot claim universality, as their own temporality will always become apparent over time and through critique.[2]

Framing positions within and across feminism through identifying specific lenses or locations applied in feminist thought has had different uses in the past. Comparisons between positions are required to identify differences. When I presented this model at a conference, I was asked directly about what place feminist indigenous perspectives might have in this model (even though indigenous perspectives are multiple within and across different tribes and ethnicities and contain different political perspectives) and, as a result, it would be difficult to identify a single unified indigenous aesthetic/political approach in feminisms. The same question can be asked about black feminist approaches, if it were possible to say that black feminism was just one thing, and not also Afro-American, Afro-Caribbean, African, queer or anti-racist in its enquiries, because there are also liberal, Marxist/left-wing/revolutionary, cosmopolitan, multi-culturalist, queer, post- and anti-colonial approaches in black feminisms and, as a result, this produces different types of work and different areas of study. In 1995, Tuzyline Jita Allan, for example, attempted to compare womanist and feminist aesthetics, playing the insights of one standpoint against another to illuminate four works of literature and show four standpoints, four kinds of subjectivities projected by black women authors in their female characters and situations (Allan 1995). The strengths and insufficiencies of womanist, black feminist and African feminist

perspectives was revealed through this method of thinking with/through specific lenses, each recognised through aesthetic models detected in named characteristics of their literary works. What this switching of comparisons between different writers could reveal was the locations from which they spoke and contrasting positions in black, American, African and European thought. As absolute distinctions in their thought were difficult to draw, Allen acknowledged that her project could only be a valuable experiment in examining the construction of certain formations of knowledge through comparative analysis.

Comparative analysis, however, does not rest on abstraction from empirical characteristics alone (linking artists by identity markers of sex, race or class, for example), nor does it rest on the subjectivity of the author, because in revealing how knowledge itself is constructed, other speculative systems are introduced. What might looking at feminist art criticism in this way contribute to re-animating debates about feminist aesthetics? While subjective reactions can be observed, measured, generalised, (following Adorno, in *Aesthetic Theory*) to do so (in art writing), was in his view, always to remain trapped in "a pre-aesthetic sphere", which actually reveals the reified ideology of the culture industry, and the problem remains that the main object of any aesthetic theory, aesthetics itself, escapes study. While he did argue "that empiricism recoils from art" and "all knowledge that does not agree with its rules of the game" can only be attributed to poetry "because art is an entity that is not identical with its empiria", he recognised that "the compulsion to aesthetics" rested on "the need to think this empirical incommensurability" (Adorno 1997, p. 426). So, if one follows Adorno, it is the incommensurability between art and its categories in philosophy or other theories of knowledge that contains the rationale in any compulsion towards aesthetics. If it were just a question of naming things correctly or classifying art objects, for example, as "feminist", then the results of any aesthetics of this kind, as Adorno suggests, "would be incomparably meagre when compared with the substantive and incisive categories of the speculative systems" (Adorno 1997, pp. 425–26). The contradictions Adorno identifies are central to the dilemma faced by any attempt to name or create a feminist aesthetics that risks either appearing meagre as the classification of a descriptive system; pre-aesthetic, if trapped in the ideology of the culture industry; or, more hopefully, resting in rethinking this incommensurability by questioning both the categories and phenomena of art as the aesthetic/politic dilemma of its own writing.

Art criticism is marked by many backlashes, often cultural amnesia, or just plain ignorance of the extended character of feminist debates and continues to marginalise women's production because it can only be marked by judgement by "Other" criteria centred around their sex, subjectivity or sexuality.[3] Woman has played this essentialising role in mainstream discourse, but feminism has sought to overcome this figure/mythic stereotype by attention to real, historical women and the details of their lives and a deconstruction of the patriarchal bias in the mythic Woman. In art criticism, no one has reached the point that women as cultural producers do not exist, (as opposed to the Lacanian formulation of the Woman does not exist), because women are still seeking representation by and on behalf of other women (in Spivak's term, a necessary and strategic essentialism!). Attention to the mythic Woman, too long associated with matters in Enlightenment thought, does not help in understanding contemporary women's position in the visual arts as it associates all women, and not the ideal Woman alone, only with physical bodily matter as a form of "nature" or "natural being" as the resource for embodied theories of knowledge. This point has been repeatedly made in feminist philosophy[4] and about aesthetic analysis of women's bodies and the metaphors associated with them, which place them as either spectacle or essence. [5]

My map (Figure 1) aimed to demonstrate that a plurality of feminist approaches exists in the present across a political/aesthetic spectrum, as well as the reality that feminism has not arrived at a final and ideal method or procedure for reading/interpreting art. Each of these four types of reading have produced distinct objects/subjects through their narratives, approach, selection of works and emphasis. Their political discourse fosters and supports different types of work–not in terms of medium or where these works are

shown but in how a subject/object is chosen and re-created in and through a set of choices. These four positions are not exhaustive options, but they are identifiable as feminisms and evident in the last 10 years. This map is not a complete system for aesthetic/political readings with the name of feminism. Identifying the method of how a subject/object was not only discussed but also "created" helps to determine the pursuit of specific tactics in the writing and, similarly, knowing the politics of certain authors led to understanding why only those methods were to be used as procedure. In identifying these preferences as aesthetic/political decisions in the methods adopted, my aim was to create a more complex guide to what has been written or attempted repeatedly and already. Inevitably, there are women writers who work across these types of approaches in different articles, which is why it is not one author's position but a demonstration of tendencies in feminist art criticism that this approach represents.

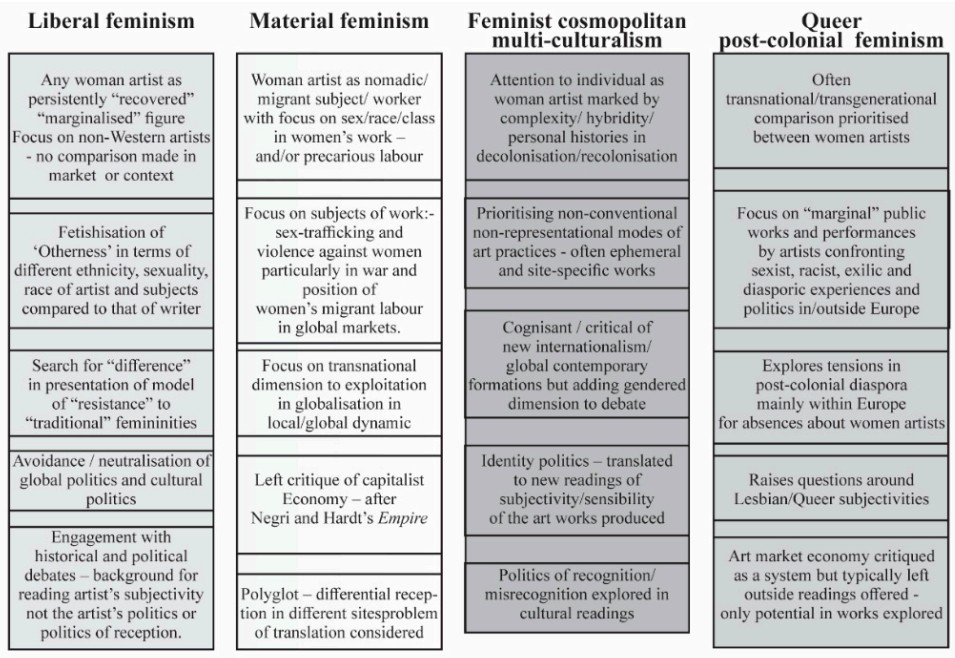

**Figure 1.** Katy Deepwell Map of 4 approaches in feminist art criticism.

The hegemonic paradigm amongst feminisms in the Eurocentric/Western-focused art world remains variants of liberal feminism. Liberal (not neo-liberal) feminism places its focus on the individual woman artist and seeks to recover their life and work by identifying achievements. The aim is to add their names to the historical account or make them role models for other women. Naming achievements has become the replacement for countering the rhetoric of biologically endowed or male-centred notions of genius, as it puts the emphasis back on women as "inventors with imagination", "pioneers" and women who have contributed to culture. As a tactic, this often dodges the question of why a special case has to be made for this woman, in particular, or what their legacy might be in other artists' works. Whether the artist is Western or non-Western, the dominant trait in these narratives is a story of recovery through careful review of a body of works or a period in their career; however, the artist's marginal status is often reinforced because her success is seen as belated and her reputation recovered only after her greatest achievements were realised. If the artists are avant-garde, this feature is often part of the process of how they gained recognition, not where a "new" case needs to be made. The artist is always presented as exceptional in terms of her life as a woman in/of her society/cultural background. This focus on the exceptional features of the artist's life or work are what makes her resistant to, different from or transgressive of traditional femininities of her time/society/culture. This exceptionalism is full of contradictions: either she did not have children or she had many children, she had women lovers or only one major artistic partner, she died early or

she continued to work for 50 years! These are all variants of not being in monogamous relationships or in a nuclear family today as these are understood implicitly as social norms in Western democracies, even when they are no longer so for most populations. In liberal feminism, there remains a fascination with, even fetishisation of, "Otherness" for people in "non-traditional" relationships and/or other societies or cultures. Fitting this "Othered" narrative as a woman is much more important to the narrative than the cultural politics of the market or global politics and whether the artist has succeeded in selling or exhibiting their work in their lifetime. There have been many critiques of this focus on individual artist's biography as the explanation for the work of art.[6] Nevertheless, it has to be said that it is these studies that have produced much of the important empirical work on women artists as names in feminist art history, dictionaries, databases and monographs: collecting and collating oeuvres, archives, and facts about women artists' lives from personal papers or relatives. Many of these accounts seek to dispel myths about women as artists with empirical information and counter-stories, but little attention is given to theoretical work or critique of the tropes of art history. Gender differences (because obviously women are "not-men", regardless of individual sexual preferences) are applied to the analysis of the subjects in the artworks studied or traits of femininity that can be detected in everything from scale, brush-stroke, detail or gesture in the work itself. Many features of modernist art criticism are retained in brief formalist analysis of the works and examination of feminine stereotypes. In identifying liberal feminism in this way, it should be said that the dominant paradigm of art history remains intact; its premises are rarely subjected to scrutiny and the binary between major and minor artists and artworks is repeated. Criteria about art itself remains unchallenged. The artistic monographic essay is a gold standard in art history but little attention is ever given in this approach to comparative analysis (between past and present or between their subject and other artists, or between competing arguments about reading a work of art).

Ryan Musgrave argued that feminism in the USA has been very dependent on a liberal legal political understanding of equality and oppression–embracing meritocracy, individualism, non-interference in the private realm and legalistic fair or sex-blind remedies in the public realm.[7] She identified two polarised approaches in how feminists have appealed to the law in discussing representations and explored their implications for aesthetics/politics in art in the USA. First, Catherine MacKinnon's conservative realism where it is the content of the work that is at issue in both misogynist hate speech and pornography, but what is required is censorship of images (largely photography and video) because these representations mimic/represent acts undertaken in real life (i.e., violence against women) and negatively manipulate audiences' emotions/expectations (perpetuating patriarchy). The only feminist antidote to this saturation of negative images would be images that are true to life, positive representations by and of women and only this. Second, Ryan Musgrave identifies a liberal realism that is the very opposite of the politics of censorship advocated by MacKinnon, where "free speech" and self-expression–a liberation–cannot be regulated by the State but must be publicly tolerated in a liberal democracy, which would be 'a liberal aesthetic that values artworks as equally valuable expressions, much like all votes' (Ibid., p. 223). In both these strategies, the content of artwork is prioritised over modernist form and: 'A certain version of mimesis, then, is at play in feminist politics of representation analyses—the conviction that it is the job of art or creative work to get it right, to show how it "really" is, to come clean of previously incorrect and ideologically weighted images' (Ibid., p. 226). She suggests that only by rethinking aesthetic theory from the point of view of radical and democratic social movements is there a route out of the dilemma posed by either a censorship in the name of liberty or libertarianism as a form of repressive tolerance.

Material feminism–in continuing a post-Marxist legacy–has sought other grounds as starting points for its analysis of women artists. Criticism is often framed around a political choice of subject matter–issues from a Left political agenda are the most frequent starting points: collective action against discrimination at work, violence against women; sex trafficking; or women's role in certain markets or workforces, including as carers or

mothers all feature strongly. Whether or not the artist discussed has had a major exhibition or a retrospective, major projects are selected–and as a result, artists who work outside the art world (as defined by dealing/commercial galleries) or the gallery as a site/situation, as well as in it, are often discussed. The focus shifts to how these issues are tackled in artworks and it moves away from individuals, paying attention to how class politics can also be read in relation to sex and race. If biography is mentioned, it is a background detail to ground an artist's identity in a social, political or economic formation and to highlight the politics of the author/artist in resisting these or questioning the status quo around them. Attempts to debate a dialectical relationship between how the artwork tackles an issue, the broader politics of the issue in society and the question of the artwork's relationship to the real (even to changing social consciousness) are the focus. The material location of the exhibition, the politics of how and when the work was exhibited are discussed much more than in liberal essays, in relation to their commercial/non-commercial positioning vis-à-vis the global art market, local scene or biennale circuits. To overcome the critique of liberal values, comparative and sometimes trans-national analysis is much stronger between artworks and artists, to demonstrate artistic or vanguardist strategies in the artist's approach or location within local/global; regional/national; national/international dynamics. Collectives of women and/or particularly collectively produced works are key subjects: especially where arguments around women in the labour market, the problems of social reproduction, women's precarity as migrant/part-time workers or second-class citizens, the politics of care and/or motherhood, and historical identifications between past and present women's struggles can be demonstrated in colonial/post-colonial and global economies. To compare projects and artists is a means to arrive at what is singular and unique about each artist or artwork. The contradictions and tensions in modern working lives for women are foregrounded, for the artist, for their subjects and in how the artworks are produced or received. As a result, thematic essays rather than one-person accounts predominate and collective actions as well as the potential for changing consciousness as a result of the encounter with the work are highlighted. Rarely does work in this group challenge the economics of the art market, unless it is a subject specific to the work itself.

In the third group, feminist cosmopolitanism–as a form of multi-culturalism–tends to foreground setting up specific contrasts between diverse women artists along racial/ethnic/religious grounds in its analyses. The construction of the essays are centred on making these types of differences apparent as forms of identification/identity between the artists or the artworks as a matter of distinctions. The themes chosen are much more diffuse and abstract than the political agendas of materialist feminisms: categories like, the home, the domestic, the uncanny, work about ecological issues/climate crisis, or abstraction, for example, are present. Whether women are studied within the confines of a nation state, or in configurations that question nationalism/regionalism or local features of a politics or an art scene, the emphasis is on how the politics of gender and race, or gender and ethnicity, or gender and religion in that region or locality informs the work. Generally, the work focuses on artists whose visibility is already secured by exhibition in well-established venues or markets, and this recognition is more important than whether the work is recognised as original in an avant garde sense or even "new". Rarely is the focus on emerging artists, tendencies or new forms of work, even as the politics of recognition and mis-recognition of the artworks are explored, because the arguments are formed in by or through discussion of the practices of well-exhibited or highly visible artists. Identity politics often grounds the choice of artists as "representative" of a nation/city, more than the subjects in the artwork, and these occasionally spill over into demonstrations of intersectionality as manifestly visible in comparisons as an identity-in-the-difference between artists or to demonstrate how each individual represents new forms of sensibility/subjectivity emerging through hybridity/diasporic politics or specific minority ethnic cultural groups, including sometimes indigenous ones. The histories of personal family backgrounds for the artist in tracing stories of post-colonialism are important here, especially where they affect the subjects chosen for artworks and how they are tackled from a personal/political perspective,

set against a global cosmopolitanism. The narratives are clearer where women artists are migrants (from South to North, or East to West), largely in the sense of a cosmopolitan elite working transnationally, even though their cross-class identification with the masses who migrate as refugees, asylum seekers or economic migrants is rarely explored, except in humanitarian terms. The agendas from cosmopolitanism, border and migration aesthetics are utilised in these articles, but gender is added to the analysis, as an extension rather than a questioning of presuppositions or criteria of other agendas.

In the fourth group, queer post-colonial feminisms, both elements–queer and post-colonial–are pronounced in who is chosen as subjects of articles. The marginality of the artist produced by these dynamics becomes central to the restorative narratives, with the aim of re-privileging certain subjects for their unique, bold or brazen qualities given their de-centering via generalised statements about Western prejudices, privileges, and Euro-centric thought. The focus is on the alternatives offered within the artworks–often as the identification factor for the writer themselves–and how the artists represent an alternative vision of the world: manifestly queer (in terms of the sexuality of artist or chosen subject matter) and/or postcolonial (in its contrast between a colonial past and a postcolonial present). Most of the artists chosen live and work in dominant cultural centres of the North or West (dependent on your viewpoint), but their exilic, marginal, critical position is also underscored by their identity–not by their appearance in the art market or at mainstream venues; even though, somewhat ironically, it is often their accommodation or acceptance there that prompts the article. Analysis of the art market, its internationalisation or its ac-commodations are left out of the analysis of the artist or their artworks, especially how the ethnic/national or critique of nationalist representations in which they engage are consid-ered. The focus is on the value of the artist in manifesting queer subjectivities/sensibilities, with or without a post-colonial twist, as a new kind of historical identity/subjectivity in the world. This identity is the work's cutting edge. The artist's activism in queer politics or their art as a representative of queerness is always registered as of primary significance for how to read the work.

What occurred to me in thinking about these four groups was how it was always completely different women artists or their artworks that were chosen by each form of analysis. Did the argument develop because of the artworks? Or were the works selected to fit the argument? It is still the case that anthologies of art criticism or monographic catalogues on artists do not generally publish more than one feminist essay, even in the name of diversity or as a representational politics, and this is why there are very few art books or magazines where you can find this full range of positions published together (*n.paradoxa* was the exception in this regard). None of these four approaches advocated a specific type of work as "feminist art", eveNotn as they offered their writing as a specific type of feminist reading. Authors rarely departed from the broader self-established liberal, materialist, post-colonial, queer or cosmopolitan frameworks and as a result seemed to reinforce their own terms of reference, rather than challenge them. Very few articles seemed to be focused on naming new tendencies or schools of thought (rejecting this as purely a modernist manner) or claiming any future predictive lines of enquiry. If a canon is supposedly arrived at (by public debate/consensus or the discipline of teaching), there was neither pedagogy or controversy here about which women artists should be subjects in feminist art criticism. While Lucy Lippard had wondered critically in the 1970s if writing about women artists was just about gaining them 'a slice of the pie', few of these articles really set out to establish the terms on which women artists could gain a place in the commercial art world, a local art scene or the global biennale circuit (working on new commissions or representing their countries in national pavilions). The search not just for a reading but a definitive preferred reading took precedence in each of these arguments; but, I wondered, do these readings really produce a change in consciousness in their readers, or is there just too neat a fit between the selection of artists and methods? Underlying my attempt to characterise these differences was the aim of working out where different futures for feminism might lie as a more radical extension of these types. I was thinking about my

own interest in developing a transnational feminism without borders (following Chandra Mohanty (Mohanty 2003)) through publishing on an internationalist platform in which the ambition was to recognise and discuss our similarities and differences, in ways which are not threatening to other perspectives, but could value the distinct contributions of others offered and at the same time succeed in creating broader alliances between feminisms and new directions for research. It is this form of thinking across different schools of thought that Chela Sandoval has also written about in her work developing a methodology of oppression (Sandoval 2000).

In 2017, I interviewed Clare Hemmings about her book, *Why Stories Matter*, which was based on analysing the "gloss" given in introductions to women's studies journal articles about the directions of feminism. She had identified in these opening statements repeated ideas about (1) the path to progress of feminism into the future, (2) a mourning for the political/cultural losses of feminism's earlier incarnations, and (3) arguments about the need for return to earlier types of feminism to move forward (Hemmings 2017; Hemmings 2011). How is feminist discourse/politics changing, developing or growing? On the one hand, it is extremely successful given the increasing volume of articles published since the 1990s self-identified and named as feminist, in exhibitions of feminist art and books on feminism, but how is it developing in terms of its own discourses and where is it in dialogue with different branches of feminisms? What are the implications of these methods for fostering only particular kinds of art/art criticism in the sense of reproduction of a type that does not recognise the existence of other forms of feminist interventions or art practices?

Adorno states: "Art is actually the world once over, as like it as it is unlike it" (Adorno 1997, p. 427). For him, "A genuine relation between art and consciousness's experience of it would consist in education, which schools opposition to art as a consumer product as much as it allows the recipient a substantial idea of what an artwork is" (Adorno 1997, p. 427). Feminist theory aimed to change consciousness, and as a result its impact in education for both art history as a discipline and art criticism as a professional practice should be transformative–if and where it was studied, reconsidered and developed, because at present, it is not a subject taught widely in universities globally. Working out a feminism that is transnational/anti-nationalist, anti-racist, embraces LGBTQI perspectives, remains critical of both capitalism and patriarchy and is not a reinforcement of Western middle-class/bourgeois values or discrimination against different ethnicities or devout religious beliefs (even as it criticises how either may discriminate against women, their sexuality or ideas of family life) is a hard task. Feminism continues to push at this hard, and often highly speculative, task for envisioning a different future, while claiming to resist reinforcing mainstream arguments and refuse a position only on the periphery. De-, anti- and post-colonial approaches have challenged how the values of the Global North are reinforced in American and Euro-Centric writing, but where are its feminisms? While the concepts of de-linking, teaching to transgress and learning to unlearn have become important, how do they coincide with long-standing ideas of ideological critique, doubled-consciousness, recognition of alternative world-views and futures from feminist thought? Isn't this what a feminist emphasis on producing readings of women artists now needs to consider?

**Funding:** This research received no external funding.

**Institutional Review Board Statement:** Not applicable.

**Informed Consent Statement:** Not applicable.

**Data Availability Statement:** Not applicable.

**Conflicts of Interest:** The author declares no conflict of interest.

## Notes

1    This chart was presented at conferences in University of Rennes (2015) and **Kjønnsforskning NÅ!**–Centre for Women's and Gender Research at UiT Norway's Arctic University, Tromso, Norway, Tromso (2019) and published in K. Deepwell 'Pourquoi 1989? Écrire sur le féminisme, l'art et le "contemporain global"/'Why 1989? Writing about feminism, art and the "global contemporary"' in Zabunyan et al. (2020).

2    Adorno (1997, p. 423). e.g., his remarks on the "obsolence of aesthetics" as a system of thought because it "scarcely confronted itself with its object" and at the same time "seemed sworn to a universality that culminates in inadequacy in the artworks, and, complementarily, in transitory eternal values".

3    'A truly genderised perspective would mean that the sex–male or female–of both the artist and the critic is taken into account. This also implies their relation to gender-values in the institutions and within the theories they apply. It cannot be stressed enough that it is impossible to deconstruct this myth of gender-neutrality in art if at the same time, male artists and critics do not develop a consciousness of their own gender.' (Ecker 1985). See also Deepwell (2020).

4    Brodribb (1992) 'The feminist project must yet elaborate an ethics and aesthetics that is not filtered through or returned to a masculine paradigm, but expressed creatively and symbolically by a subject that is female. Only an unflinching autonomy can challenge extortions to feminine deference and the deferment of feminist philosophy. Women's memory is annulled in the patriarchal tradition . . . .The postmodern male author is dying, imploding as a subject, while women are claiming a voice, giving birth to a feminist movement and vision and remembering against dismemberment . . . Our knowledge is untranslatable and inaudible in mixed forums of masculine hegemony. Yet *les homes roses* abstract and parade a feminist language and theory made textureless, without body, without speaking, female bodies. We serve as raw matter for an unaltered analysis which has none of our values, we do not control this speech, insidious, neutralised, dishonest recognition of the female, spoken in a sexist practice'.

5    Gisela Ecker introduction to *Feminist* Aesthetics (1985) 'It is tempting to mistake features of women's art as representations of 'women's nature'. The opinions that revolve around the body in contemporary art by women may serve to illustrate this. Menstrual blood, clitoral images, feminine body language and pregnancy are used to bring into play aspects of female sexuality that are absent or even repressed in male art, yet this iconography of the body is not displayed in the sense of 'how women are' but employed by women artists who have a political consciousness of sexual difference in art . . . .Because art is open to multiple interpretation, the old critical categories (i.e., the ideologies which created them) are extremely persistent. Critics still manage to offer explanations in which they continue to see women as a spectacle and essence rather than recognising the function of these performances as a process and an artificial construct.' pp. 17–18.

6    Pollock (1982), Gouma-Peterson and Mathews (1987) and many other essays. Including: (Deepwell 2022).

7    Musgrave (2003). *JSTOR*, http://www.jstor.org/stable/3810981, accessed on 13 March 2023. On p. 233, on p. 232 she names examples of a critical feminist aesthetics in: Winterson's *Art Objects: Essays on Ecstasy and Effrontery*; Brand's collection *Beauty* Matters (2000); Isobel Armstrong's *The Radical* Aesthetic (2000); Florence and Foster's collection *Differential Aesthetics: Art Practices, Philosophy and Feminist* Understandings (2000); Cynthia Freeland's *But Is It* Art? (2001); and Maggie O'Neill's edited collection, *Adorno,* Culture and Feminism (1999) as demonstrating this shift.

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
