# Peer review of "The Politics and Aesthetic Choices of Feminist Art Criticism"

_arts, 2022_

Round 1

Reviewer 1 Report

This is an interesting paper, offering innovative approach to the feminist aesthetics and its main currents. The preoccupation with the diversity within feminist theory is the article's main strength, as is the ability to discuss the postcolonial and transnational dimensions of feminism and art. As a grid organizing the feminist aesthetic field, the analysis offered in this article could be very helpful, and allow a genuinely inclusive art theorizing from the feminist standpoint. However, some changes and additions could perhaps be made in order to better situate it within feminist discussions of aesthetics. The authors from Eastern Europe also discuss the regional divisions, like Agata Jakubowska, Izabela Kowalczyk, Marina Griznic or Boyana Kunst, maybe it would be worth looking into their work? There are some authors formulating central elements of feminist aesthetics, like Griselda Pollock, Linda Nochlin and others, perhaps engaging with them could make the argumentation of this article somewhat better situated in the field? Also - Adorno is a very important author of theory of aesthetics relevant to the discussion offered in the article, however, his is definitely not the only perspective on post-war aesthetics that was created, and definitely not the newest, perhaps Hal Foster's theory of neo-avant-garde is also a good reference, as the other authors the article engages with are much more contemporary? This is maybe to be considered - how to relate to newer aesthetic theory. I would also change the abstract, in order to make it more specific and thus to give a better, more precise idea as to what and how the Author would like to proceed. But - although I have many suggestions, I believe this article offers a relevant, contemporary approach to feminist aesthetics, and allows situating transnational and decolonial perspectives in it, which is a great opportunity for the future developments of the discipline. 

Author Response

I wrote very specifically that I did not offer names for each of the tendencies I identified. I've worked and published many of the feminist authors listed by the reviewer (with the exception of Kunst and Nochlin) in n.paradoxa and the feminist articles reflected upon are the result of being a commissioning editor and reading widely in the field.

Adorno is used as a reflective foil in the argument to make specific points. It is not an Adorno-based analysis. The argument could equally have returned to Foster, or Greenberg, Rosenberg, Krauss, Owens amongst many others to emphasise a North American lineage, but this is not the purpose of the article - nor was it intended to give a potted history of avantgardism in this article.

Similarly to write a critique of new aesthetic theory, I could have chosen many other authors, Mieke Bal, Craig Leonard, Peter Osborne to engage with.... and it would be another book, not an article.

I have amended the abstract.

I've done some minor copyediting of the draft and small adjustments.

Reviewer 2 Report

l. 51 claim (not claims)

L. 97. semi-colon 'but set against these;

 * Note 6, which non-francophoone reviewers will not perceive as incorrect:

 Deepwell article title - add accents:

'Écrire sur le féminisme.....

Book title - lower case and accents:

et al., (NB full stop before comma) Constellations subjectives : pour une histoire féministe de l’art (France, Éditions iXe, 2020)

Author Response

I've attended to the recommended copyedits.